# Factors Associated with Reduced Heart Rate Variability in the General Japanese Population: The Iwaki Cross-Sectional Research Study

**DOI:** 10.3390/healthcare10050793

**Published:** 2022-04-24

**Authors:** Masaya Tsubokawa, Miyuki Nishimura, Yoshinori Tamada, Shigeyuki Nakaji

**Affiliations:** 1Innovation Center for Health Promotion, Graduate School of Medicine, Hirosaki University, Hirosaki 036-8562, Japan; y.tamada@hirosaki-u.ac.jp (Y.T.); nakaji@hirosaki-u.ac.jp (S.N.); 2Health Science Research Center, FANCL Research Institute, Yokohama 244-0806, Japan; nishimura_miyuki@fancl.co.jp

**Keywords:** general Japanese population, heart rate variability, plasma pentosidine, R–R interval

## Abstract

Although many studies have reported factors associated with reduced heart rate variability (HRV) in Western populations, evidence is limited among Asian populations. Therefore, we investigated the factors associated with reduced HRV values in a general Japanese population by measuring HRV among the participants of the Iwaki Health Promotion Project who underwent medical examination in 2019. We performed 90-s HRV measurements in 1065 participants. Of these, we evaluated the coefficient of variation in R–R intervals (CVRR) and standard deviation in R–R intervals (SDNN). Blood was collected under a fasting condition, and investigations of glucose metabolism, lipid metabolism, renal function, liver function, advanced glycation end products, and blood pressure were performed. A multivariate regression analysis of the association between CVRR or SDNN and blood test parameters and blood pressure in 987 participants with adequately completed HRV assessments showed that reduced CVRR or SDNN was associated with higher levels of glycated hemoglobin (HbA1c), glycoalbumin, blood glucose, triglycerides, creatinine, plasma pentosidine, and diastolic blood pressure. In the general Japanese population, higher levels of HbA1c, glycoalbumin, blood glucose, triglycerides, creatinine, plasma pentosidine, and diastolic blood pressure are associated with reduced CVRR or SDNN, which are typical HRV parameters.

## 1. Introduction

The autonomic nervous system is part of the peripheral nervous system and regulates involuntary physiological processes such as heart rate, blood pressure, respiration, digestion, and sexual arousal [1]. It is also integral in maintaining homeostasis and behavioral functions [2]. Autonomic nervous system dysfunction is a known complication of diabetes mellitus [3,4], leading to diabetic autonomic neuropathies, including tachycardia at rest, exercise intolerance, orthostatic hypotension, constipation, gastroparesis, erectile dysfunction, hypoglycemia unawareness, and disordered perspiration [5].

To clinically evaluate the autonomic nervous function, a heart rate variability (HRV) analysis, an index of the autonomic nervous activity related to cardiovascular function, has been applied as a non-invasive measurement in various fields [6,7]. Reduced HRV has been identified in patients with diabetes [8,9], reportedly associated with an autonomic symptom score [10], and it is an important factor associated with an. increased risk of coronary heart disease (CHD) [11] and death [12,13]. Reduced HRV in patients with myocardial infarction is also associated with an increased risk of death [14].

In the general population, HRV is clinically significant as its reduction reportedly increases the risk of cardiovascular diseases (CVD) [15] and is associated with frailty [16]. Frailty is characterized by a decline in the physiological function of older adults, thereby increasing their vulnerability to stressors [17,18]. Additionally, frailty increases the risk of reduced quality of life [19], deteriorated activities of daily living, hospital admission, and mortality [20,21]. Hence, to prevent CVD and frailty and extend a healthy life span, it is crucial to investigate the factors associated with reduced HRV in the general population, to enable the necessary measures against reduced HRV.

Although many studies have investigated the factors associated with reduced HRV in Western populations, reports in Asia are limited [22]. Therefore, we investigated the factors associated with reduced HRV in the Japanese general population by assessing the HRV of the participants of the Iwaki Health Promotion Project who underwent a medical examination.

## 2. Materials and Methods

### 2.1. Participants and Analysis

The Iwaki Health Promotion Project medical examination has been conducted annually since 2005, with the aim of preventing lifestyle-related diseases, maintaining and promoting health, and extending the life span of residents of the Iwaki District, Hirosaki City [23]. Participants of the 2019 medical examination (*n* = 1065) were recruited from among men and women aged ≥20 years living in Iwaki District, Hirosaki City, Aomori Prefecture. Among them, this cross-sectional study targeted 987 participants who completed HRV measurements. This study was approved by the Ethics Review Board of Hirosaki University School of Medicine (approval number: 2019-009) and was conducted according to the principles of the Declaration of Helsinki. Written informed consent was obtained from all participants.

### 2.2. Clinical Features

All laboratory tests were performed under fasting conditions in the morning. Body mass index (BMI) was assessed during the physical examination; systolic blood pressure (SBP) and diastolic blood pressure (DBP) were also measured. BMI was calculated as follows: BMI = weight (kg)/height (m)^2^. Blood pressure was measured using an automated sphygmomanometer Elemano2 (Terumo Co., Ltd., Tokyo, Japan), with the participants at rest while sitting. Blood tests included the levels of hemoglobin A1c (HbA1c), glycoalbumin, and blood glucose to assess glucose metabolic capacity; triglyceride, total cholesterol, high-density lipoprotein (HDL), and low-density lipoprotein (LDL) to assess lipid metabolic capacity; alanine transaminase (ALT), aspartate transaminase (AST), and γ-glutamyl transferase (γGTP) to assess liver function; creatinine and urea nitrogen to assess renal function; and plasma pentosidine to assess advanced glycation end products (AGEs). Blood samples were collected from the peripheral veins, with the participants lying supine, and blood tests were performed by LSI Medience Co., Ltd. (Tokyo, Japan). To analyze lifestyle-related factors, we retrieved the survey results for diabetes mellitus, dyslipidemia, hypertension, smoking, drinking, exercise (non-winter and winter seasons), and antihypertensive use.

### 2.3. Measurement of HRV

HRV was also measured under fasting conditions in the morning. The participants did not smoke or perform vigorous activities during the health examination. Using a Vital Monitor 302 (VM302) system (Hitachi Systems, Inc., Tokyo, Japan), 90-s measurements were performed, with the participants seated with their eyes closed and at rest after ensuring that the heartbeat was stable. The data were analyzed using Memfmcc software (Institute of Fatigue Sciences, Osaka, Japan). VM302 can simultaneously conduct electrocardiography (ECG) and photoplethysmography (PPG) from fingertips and has been utilized in several clinical trials [24,25]. By monitoring HRV using ECG and PPG, we collected the 90-s cardiac autonomic nervous function data, at a sampling rate of 600 Hz. The built-in firmware in VM302 detected the R-wave peaks using a peak detection algorithm based on a hill-climbing method [26], and the obtained peak time on the R waves was transmitted to an external computer. From each R-peak time, a time series of R–R intervals was generated sequentially. The noise of the digital signals was removed using a low-pass filter [27,28], whereas the linear trends for the R–R intervals were not removed. Heart rate (HR) was calculated from the inverse of the R–R intervals in each heartbeat. Additionally, instead of the stationary test for R–R interval fluctuations, errors in the 30-s data of R–R intervals were defined as values that were 0.75-fold lower and 1.75-fold higher than the median values of R–R intervals for 30 s, and the matched data of R–R intervals were removed. We excluded 78 participants with heart rate instability errors or no HRV measurements using the VM302.

We evaluated the coefficient of variation of R–R intervals (CVRR) and standard deviation in R–R intervals (SDNN), which represent the modulations in both sympathetic and parasympathetic functions [29] as time domain parameters, which are considered to be relatively accurate even for short durations [30]. SDNN and CVRR were calculated as the standard deviation of the R–R interval and SDNN/mean value of the R–R interval × 100, respectively, during the measurement. Although thresholds for SDNN and CVRR have been reported for mortality in patients with depressed left ventricular function after an acute myocardial infarction and assessment of diabetic autonomic neuropathy [31,32], we currently do not have sufficient information to identify the cutoff values for the HRV indices [33]. Therefore, in this study, they were compared as continuous parameters. A frequency analysis was performed for reference. The low-frequency component power (LF) was calculated as the power within a frequency range of 0.04–0.15 Hz, and the high-frequency component power (HF) was calculated as the power within a frequency range of 0.15–0.4 Hz. The mean values of LF, HF, and LF/HF obtained in each time series were considered as representative values for each measurement. Frequency analyses for R–R interval variation were performed using the maximum entropy method, which can estimate the power spectrum density from short time-series data and is adequate for examining changes in HRV under different conditions of short durations.

### 2.4. Statistical Analysis

Among the clinical characteristics of the participants, continuous variables were presented as means and standard deviations (SD), whereas categorical variables were presented as sample size and percentage (%) as summary statistics for each variable. To investigate the factors associated with HRV, a univariate regression analysis was performed, with CVRR and SDNN as the objective variables and age, sex, BMI, individual hematology parameters, and blood pressure measurements as explanatory variables. Additionally, multivariate regression analyses, adjusted for age, sex, and BMI (Model 1) and adjusted for covariates in Model 1 plus smoking, drinking, exercise (non-winter and winter seasons), and antihypertensive use (Model 2), were performed. For reference, a frequency analysis (LF, HF, LF/HF) was conducted using the same method. The regression coefficient (β) and its 95% confidence interval (CI) were calculated, with the level of significance set at *p* < 0.05 in two-tailed tests. All statistical analyses were conducted using the JMP^®^ 15 software (SAS Institute Inc., Cary, NC, USA), and the regression analysis was performed using the standard least-squares method.

## 3. Results

Table 1 shows the clinical characteristics of the study participants. The characteristics of the study cohort were as follows: sex distribution, 59.0% women and 41.0% men; mean age, 52.3 years; and mean BMI, 23.1 kg/m^2^.

Table 2, Table 3 and Table 4 show the results of the univariate and multivariate regression analysis of the association between CVRR or SDNN and laboratory data.

The statistically significant associations by the univariate regression analysis with SDNN or CVRR were discerned with age, BMI, HbA1c, glycoalbumin, blood glucose, triglyceride, total cholesterol, LDL-cholesterol, AST, γ-GTP, urea nitrogen, creatinine, plasma pentosidine, SBP, and DBP (Table 2), but total cholesterol, LDL-cholesterol, AST, γ-GTP, urea nitrogen, and SBP were marginal clinical significance because these could not be confirmed by the multivariate analysis (Table 3 and Table 4). 

Among those with significant differences in multivariate regression analysis adjusted for age, sex, and BMI, CVRR decreased by 0.174%, 0.055%, 0.002%, 0.185%, 0.006%, and 0.013% per unit increase in HbA1c, glycoalbumin, triglyceride, creatinine, plasma pentosidine, and DBP, respectively (Table 3). Additionally, among those with significant differences in the same multivariate regression analysis (model 1), SDNN decreased by 1.925 ms, 0.505 ms, 0.086 ms, 0.019 ms, 1.682 ms, 0.061 ms, and 0.173 ms per unit increase in HbA1c, glycoalbumin, blood glucose, triglyceride, creatinine, plasma pentosidine, and DBP, respectively (Table 3). The results were similar in the multivariate regression analyses adjusted for smoking, drinking, exercise (non-winter and winter seasons), and antihypertensive use, in addition to age, sex, and BMI (Table 4). The important associations by the multivariate regression analysis with SDNN or CVRR were between HbA1c, glycoalbumin, blood glucose, triglyceride, creatinine, plasma pentosidine, or DBP.

The results of the frequency analysis (LF, HF, and LF/HF) are given in Appendix A. The multivariate regression analysis identified associations between decreased LF and higher levels of DBP, as well as decreased HF and higher levels of blood glucose, triglyceride, and DBP.

## 4. Discussion

Among the general Japanese population that participated in the Iwaki Health Promotion Project medical examination, reduced CVRR or SDNN, a typical measure of HRV, was associated with increased HbA1c, glycoalbumin, blood glucose, triglyceride, creatinine, plasma pentosidine levels, and DBP. Interestingly, the association between reduced HRV and high plasma pentosidine levels has not been reported previously.

Regarding the association between reduced CVRR or SDNN and the deterioration of glyco-metabolism, high blood glucose levels may lead to neurological dysfunction through multiple mechanisms, such as the induction of oxidative stress and toxic glycosylation products [34,35]. High HbA1c levels reportedly reduce HRV in people with diabetes [36]; therefore, high HbA1c levels may also have influenced the reduced HRV in the general population in this study. However, high glycoalbumin levels reflect the blood glucose status for a shorter period than HbA1c and are less affected by health conditions such as anemia, renal failure, and pregnancy, which reduces the reliability of HbA1c measurement [37,38]. Similar to HbA1c, in the present results, high glycoalbumin levels were associated with reduced CVRR and SDNN. With respect to the association between reduced CVRR or SDNN and high triglyceride (TG) levels, increased oxidized LDL in patients with dyslipidemia may cause neurological dysfunction [39]. High TG levels reportedly reduce HRV in patients with diabetes [40]. Similar to high blood glucose levels, high TG levels may also cause reduced HRV in the general population. However, the multivariate analysis in the present study showed no association between reduced CVRR or SDNN and high LDL levels, emphasizing the need for further investigation of the underlying mechanism. Furthermore, only DBP, and not SBP, was associated with reduced CVRR and SDNN in this study. Previous studies in Japan have reported that decreased cardiac autonomic function is associated with increased blood pressure, and HRV strongly reflects DBP. A reason may be the influence of alcohol consumption on SBP levels, which weakens the association with HRV, especially in men [41]. High blood pressure is a cause of autonomic neuropathy in diabetes [42]. Moreover, reduced HRV triggers hypertension [30], the mechanism for which may involve reduced parasympathetic function and sympathetic overactivity [43]. Therefore, reduced HRV may be caused by high blood pressure and negatively affect high blood pressure. High blood glucose and TG levels and high blood pressure are components of metabolic syndrome [44]. Therefore, the prevention of high TG and blood pressure in addition to high blood glucose levels may be important, as metabolic syndrome is reportedly associated with HRV reduction in both Western and Japanese populations [45,46].

Plasma pentosidine is a typical glycation marker of AGEs [47,48], and the accumulation of AGEs is believed to induce inflammation and abnormal signaling in cells, negatively affecting neurological function [49,50]. Generally, obesity reduces HRV [51,52], and obesity in people with diabetes is also associated with reduced HRV [53]. In addition, high blood glucose in obese individuals was shown to be associated with reduced HRV [54]. Contrarily, plasma pentosidine levels are inversely associated with BMI [55,56]. Elevated plasma pentosidine levels may reduce HRV through an obesity-independent mechanism. Plasma pentosidine is also a known marker of renal function. The results of the present study confirmed the association between reduced CVRR or SDNN and high creatinine levels; thus, reduced renal function may be related to reduced HRV [57,58]. Previous reports have suggested the associations of reduced HRV with reduced renal function and status as a complication of chronic kidney disease (CKD) [59,60]. The mechanism for reduced HRV and renal function involves activation of the renin–angiotensin system [61,62]. Nonetheless, reduced HRV may also be a risk factor for CKD [63]. Therefore, further studies need to clarify the causal association between reduced HRV and renal function.

Herein, we could not confirm the association between reduced CVRR or SDNN and elevated levels of liver function markers. Although an association between nonalcoholic cirrhosis and reduced HRV has been reported [64], evidence is lacking on the association between liver function markers and HRV in the general population, making further exploration necessary.

The present findings and previous reports in both Western and Japanese populations demonstrate that HRV decreases with age and that the measurement of HRV could be a simple anti-aging test [65,66,67]. Other than preventing elevations in blood glucose, TG, and blood pressure to suppress HRV, measures to reduce AGE accumulation, such as plasma pentosidine, could reduce the risk of CVD and prevent frailty.

This study had several limitations. First, the HRV measurement in this study was not optimal. Generally, HRV should be measured for 24 h or >5 min with individuals lying supine [68]; however, in this study, the measurements were performed with the participants in a seated position for 90 s, which is a relatively short duration. Hence, some individuals with an unstable heart rate or those without HRV measurements were excluded from the analysis, which may have introduced bias in the results. Furthermore, HRV parameters were characterized by large random variations within individuals, suggesting low absolute reliability of the short-term measurements in this study. We cannot deny the possibility that circadian variations may have affected these associations. Second, although we adjusted for antihypertensive use, the effects of specific classes of antihypertensives that may be related to cardiac autonomic function could not be completely surveyed. Third, due to cross-sectional design, the causal relationship between the factors associated with reduced HRV could not be determined. In particular, the association between reduced HRV and AGE accumulation has not been previously reported, and the relationship between reduced HRV and reduced renal function does not have consistent evidence. However, the causal relationships of reduced HRV with AGE accumulation and reduced renal function can only be explored using longitudinal data. Fourth, this study included participants who voluntarily participated in the Iwaki Health Promotion Medical Examination. Consequently, it may not be possible to generalize the results of this study because of selection bias.

Additional studies are needed to clarify the causal relationships of reduced HRV with AGE accumulation and reduced renal function in the general population by obtaining longitudinal data in an appropriate measurement environment and large sample size.

## 5. Conclusions

In the Japanese general population, higher levels of HbA1c, glycoalbumin, blood glucose, triglycerides, creatinine, plasma pentosidine, and diastolic blood pressure were associated with reduced CVRR or SDNN, which is a typical measure of HRV parameters.

## Figures and Tables

**Table 1 healthcare-10-00793-t001:** Population characteristics.

		Total	(N = 987)	Men	(N = 405)	Women	(N = 582)
Continuous variables	Unit	Mean	(SD)	Mean	(SD)	Mean	(SD)
Age	years	52.3	(15.0)	52.2	(14.9)	52.4	(15.1)
BMI	kg/m^2^	23.1	(3.6)	24.1	(3.5)	22.4	(3.5)
HbA1c	%	5.7	(0.6)	5.7	(0.7)	5.7	(0.5)
Glycoalbumin	%	14.6	(2.0)	14.4	(2.4)	14.7	(1.6)
Blood glucose	mg/dL	96.1	(16.3)	99.5	(18.1)	93.6	(14.5)
Triglyceride	mg/dL	97.9	(83.7)	124.8	(113.7)	79.2	(45.1)
Total cholesterol	mg/dL	204.7	(34.7)	202.3	(34.6)	206.4	(34.8)
HDL cholesterol	mg/dL	64.8	(16.6)	58.0	(14.9)	69.6	(16.1)
LDL cholesterol	mg/dL	116.4	(30.1)	116.9	(29.5)	116.1	(30.5)
ALT	U/L	21.0	(14.2)	26.7	(17.2)	16.9	(9.8)
AST	U/L	21.8	(7.9)	23.8	(8.4)	20.4	(7.3)
γ-GTP	U/L	33.1	(40.8)	48.9	(56.6)	22.2	(17.7)
Creatinine	mg/dL	0.7	(0.5)	0.9	(0.7)	0.6	(0.3)
Urea nitrogen	mg/dL	14.5	(4.5)	15.3	(4.5)	13.9	(4.4)
Plasma pentosidine	pmol/mL	26.5	(16.5)	26.2	(19.2)	26.8	(14.4)
SBP	mmHg	120.6	(16.9)	123.6	(16.8)	118.6	(16.7)
DBP	mmHg	76.9	(11.3)	79.6	(11.6)	75.0	(10.7)
Heart rate	bpm	70.1	(10.1)	68.7	(10.2)	71.1	(9.9)
CVRR	%	3.4	(1.6)	3.4	(1.7)	3.4	(1.6)
SDNN	ms	29.5	(14.6)	30.0	(15.7)	29.1	(13.8)
LF	ms^2^	345.7	(519.5)	427.0	(624.7)	289.1	(422.9)
HF	ms^2^	256.4	(343.7)	249.3	(363.3)	261.2	(329.7)
LF/HF	Ratio	2.7	(4.6)	3.3	(4.8)	2.3	(4.4)
Categorical variables		N	(%)	N	(%)	N	(%)
Diabetes mellitus	No	926	(94.0)	374	(92.3)	552	(95.2)
	Yes	59	(6.0)	31	(7.7)	28	(4.8)
Hyperlipidemia	No	812	(82.6)	328	(81.4)	484	(83.4)
	Yes	171	(17.4)	75	(18.6)	96	(16.6)
High blood pressure	No	739	(74.9)	283	(69.9)	456	(78.5)
	Yes	247	(25.1)	122	(30.1)	125	(21.5)
Use of antihypertensive medication	No	757	(76.7)	294	(72.6)	463	(79.6)
	Yes	230	(23.3)	111	(27.4)	119	(20.4)
Exercising (except in winter)	No	761	(77.4)	311	(77.2)	450	(77.6)
	Yes	222	(22.6)	92	(22.8)	130	(22.4)
Exercising (winter)	No	759	(77.7)	314	(78.3)	445	(77.3)
	Yes	218	(22.3)	87	(21.7)	131	(22.7)
Smoking	No	616	(62.9)	166	(41.4)	450	(77.9)
	Current	173	(17.7)	121	(30.2)	52	(9.0)
	Previous	190	(19.4)	114	(28.4)	76	(13.1)
Alcohol consumption	No	464	(47.6)	116	(28.9)	348	(60.7)
	Current	471	(48.3)	275	(68.4)	196	(34.2)
	Previous	40	(4.1)	11	(2.7)	29	(5.1)

γ-GTP, γ-glutamyl transferase; ALT, alanine transaminase; AST, aspartate transaminase; BMI, body mass index; bpm, beats per minute; CVRR, coefficient of variation of R–R intervals; DBP, diastolic blood pressure; HbA1c, hemoglobin A1c; HDL, high density lipoprotein; HF, high-frequency component power; LDL, low density lipoprotein; LF, low-frequency component power; SBP, systolic blood pressure; SD, standard deviation; SDNN, standard deviation in R–R intervals.

**Table 2 healthcare-10-00793-t002:** Univariate analysis of association with CVRR and SDNN.

		CVRR (%)					SDNN (ms)				
Characteristics	Unit	β	95% CI			*p*-Value	β	95% CI			*p*-Value
Age	years	−0.046	−0.052	~	−0.040	<0.001	−0.345	−0.402	~	−0.288	<0.001
Sex	women	0.039	−0.167	~	0.245	0.710	−0.853	−2.705	~	0.998	0.366
BMI	kg/m^2^	−0.069	−0.096	~	−0.041	<0.001	−0.533	−0.782	~	−0.284	<0.001
HbA1c	%	−0.548	−0.710	~	−0.386	<0.001	−4.659	−6.122	~	−3.196	<0.001
Glycoalbumin	%	−0.152	−0.203	~	−0.101	<0.001	−1.229	−1.687	~	−0.771	<0.001
Blood glucose	mg/dL	−0.021	−0.027	~	−0.015	<0.001	−0.190	−0.245	~	−0.136	<0.001
Triglyceride	mg/dL	−0.002	−0.003	~	−0.001	<0.001	−0.021	−0.032	~	−0.010	<0.001
Total cholesterol	mg/dL	−0.005	−0.008	~	−0.002	<0.001	−0.038	−0.065	~	−0.012	0.004
HDL cholesterol	mg/dL	0.000	−0.006	~	0.006	0.929	0.016	−0.039	~	0.071	0.577
LDL cholesterol	mg/dL	−0.005	−0.008	~	−0.001	0.007	−0.032	−0.062	~	−0.002	0.036
ALT	U/L	−0.004	−0.011	~	0.003	0.249	−0.042	−0.107	~	0.022	0.198
AST	U/L	−0.024	−0.037	~	−0.011	<0.001	−0.187	−0.302	~	−0.073	0.001
γ-GTP	U/L	−0.003	−0.005	~	−0.001	0.019	−0.023	−0.046	~	−0.001	0.040
Creatinine	mg/dL	−0.236	−0.424	~	−0.049	0.014	−1.837	−3.525	~	−0.148	0.033
Urea nitrogen	mg/dL	−0.078	−0.100	~	−0.056	<0.001	−0.585	−0.785	~	−0.386	<0.001
Plasma pentosidine	pmol/mL	−0.015	−0.021	~	−0.009	<0.001	−0.129	−0.184	~	−0.075	<0.001
SBP	mmHg	−0.019	−0.025	~	−0.014	<0.001	−0.158	−0.211	~	−0.105	<0.001
DBP	mmHg	−0.027	−0.035	~	−0.018	<0.001	−0.262	−0.340	~	−0.183	<0.001

γ-GTP, γ-glutamyl transferase; ALT, alanine transaminase; AST, aspartate transaminase; BMI, body mass index; CI, confidence interval; CVRR, coefficient of variation of R–R intervals; DBP, diastolic blood pressure; HbA1c, hemoglobin A1c; HDL, high density lipoprotein; LDL, low density lipoprotein; SBP, systolic blood pressure; SDNN, standard deviation in R–R intervals.

**Table 3 healthcare-10-00793-t003:** Multivariate analysis of association with CVRR and SDNN (Model 1).

		CVRR (%)					SDNN (ms)				
Characteristics	Unit	β	95% CI			*p*-Value	β	95% CI			*p*-Value
HbA1c	%	−0.174	−0.335	~	−0.012	0.035	−1.925	−3.426	~	−0.423	0.012
Glycoalbumin	%	−0.055	−0.104	~	−0.006	0.027	−0.505	−0.961	~	−0.049	0.030
Blood glucose	mg/dL	−0.006	−0.012	~	0.001	0.074	−0.086	−0.144	~	−0.028	0.004
Triglyceride	mg/dL	−0.002	−0.003	~	−0.001	0.004	−0.019	−0.029	~	−0.008	0.001
Total cholesterol	mg/dL	0.000	−0.003	~	0.002	0.847	0.001	−0.024	~	0.027	0.920
HDL cholesterol	mg/dL	−0.001	−0.007	~	0.005	0.704	0.018	−0.040	~	0.076	0.541
LDL cholesterol	mg/dL	0.001	−0.002	~	0.004	0.453	0.013	−0.017	~	0.042	0.397
ALT	U/L	−0.001	−0.008	~	0.006	0.772	−0.033	−0.102	~	0.036	0.344
AST	U/L	−0.003	−0.015	~	0.009	0.661	−0.042	−0.156	~	0.072	0.469
γ-GTP	U/L	−0.002	−0.004	~	0.001	0.128	−0.020	−0.042	~	0.003	0.082
Creatinine	mg/dL	−0.185	−0.357	~	−0.012	0.036	−1.682	−3.291	~	−0.073	0.040
Urea nitrogen	mg/dL	−0.019	−0.041	~	0.003	0.093	−0.167	−0.375	~	0.041	0.115
Plasma pentosidine	pmol/mL	−0.006	−0.012	~	−0.00003	0.049	−0.061	−0.114	~	−0.007	0.026
SBP	mmHg	−0.003	−0.009	~	0.003	0.283	−0.044	−0.099	~	0.012	0.125
DBP	mmHg	−0.013	−0.021	~	−0.004	0.004	−0.173	−0.252	~	−0.093	<0.001

Model 1: Adjusted for age, sex, and BMI. γ-GTP, γ-glutamyl transferase; ALT, alanine transaminase; AST, aspartate transaminase; BMI, body mass index; CI, confidence interval; CVRR, coefficient of variation of R–R intervals; DBP, diastolic blood pressure; HbA1c, hemoglobin A1c; HDL, high density lipoprotein; LDL, low density lipoprotein; SBP, systolic blood pressure; SDNN, standard deviation in R–R intervals.

**Table 4 healthcare-10-00793-t004:** Multivariate analysis of association with CVRR and SDNN (Model 2).

		CVRR (%)					SDNN (ms)				
Characteristics	Unit	β	95% CI			*p*-Value	β	95% CI			*p*-Value
HbA1c	%	−0.175	−0.341	~	−0.010	0.038	−1.878	−3.416	~	−0.339	0.017
Glycoalbumin	%	−0.054	−0.104	~	−0.004	0.034	−0.481	−0.945	~	−0.017	0.042
Blood glucose	mg/dL	−0.006	−0.012	~	0.001	0.077	−0.084	−0.144	~	−0.025	0.006
Triglyceride	mg/dL	−0.002	−0.003	~	−0.001	0.005	−0.018	−0.029	~	−0.007	0.001
Total cholesterol	mg/dL	0.000	−0.003	~	0.003	0.929	0.002	−0.024	~	0.028	0.887
HDL cholesterol	mg/dL	−0.001	−0.007	~	0.006	0.802	0.018	−0.043	~	0.079	0.556
LDL cholesterol	mg/dL	0.001	−0.002	~	0.005	0.430	0.013	−0.017	~	0.044	0.384
ALT	U/L	−0.001	−0.009	~	0.006	0.708	−0.033	−0.103	~	0.037	0.360
AST	U/L	−0.004	−0.016	~	0.009	0.564	−0.048	−0.164	~	0.068	0.419
γ-GTP	U/L	−0.002	−0.004	~	0.001	0.165	−0.019	−0.041	~	0.004	0.106
Creatinine	mg/dL	−0.187	−0.364	~	−0.011	0.037	−1.675	−3.316	~	−0.034	0.045
Urea nitrogen	mg/dL	−0.024	−0.047	~	0.000	0.051	−0.199	−0.419	~	0.021	0.076
Plasma pentosidine	pmol/mL	−0.006	−0.012	~	−0.00008	0.047	−0.060	−0.117	~	−0.003	0.039
SBP	mmHg	−0.003	−0.009	~	0.003	0.354	−0.040	−0.097	~	0.018	0.176
DBP	mmHg	−0.012	−0.021	~	−0.003	0.007	−0.173	−0.255	~	−0.091	<0.001

Model 2: Adjusted for age, sex, BMI, antihypertensive medication use, physical activity (non-winter and winter seasons), smoking, and alcohol drinking. γ-GTP, γ-glutamyl transferase; ALT, alanine transaminase; AST, aspartate transaminase; BMI, body mass index; CI, confidence interval; CVRR, coefficient of variation of R–R intervals; DBP, diastolic blood pressure; HbA1c, hemoglobin A1c; HDL, high density lipoprotein; LDL, low density lipoprotein; SBP, systolic blood pressure; SDNN, standard deviation in R–R intervals.

## Data Availability

The data cannot be shared publicly because of ethical concerns. Data are available from the Hirosaki University COI Program Institutional Data Access/Ethics Committee (contact via e-mail: coi@hirosaki-u.ac.jp) for researchers who meet the criteria for access to the data. Researchers need to have prior approval from the research ethics review boards of their respective affiliations.

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
