# Peer review of "Factors Associated with Reduced Heart Rate Variability in the General Japanese Population: The Iwaki Cross-Sectional Research Study"

_healthcare, 2022, doi:10.3390/healthcare10050793_

Round 1
Reviewer 1 Report
The paper by Tsubokawa and co-authors is clearly written and informative. The presentation of results is also good. I only have some minor suggestions for it.
While the paper is novel, some additional references that deal with heart rate variability among Japanese populations should be included in this paper and discussed. For example papers like DOI: 10.2114/jpa2.26.173 and DOI: 10.1038/hr.2014.73.
Also, it is mentioned that many studies like this one in Western populations are performed. Discussion about how results obtained in this study correlate with those on Western populations should be added. If there are any differences, then what could be the reasons for those.
Line 13 in the abstract: word general is mentioned two times. The second one should be removed.
Author Response
- We have added citations to studies of HRV in Japan, including the references you provided.
- Metabolic syndrome is reportedly associated with HRV reduction in both Westerners and Japanese.
- The previous reports in both Westerners and Japanese demonstrate that HRV decreases with age.
Reviewer 2 Report
The paper studied the association of short-term heart rate variability with a number of demographic and laboratory parameters in a general Japanese population. The authors showed a significant correlation between basic HRV parameters and higher levels of HbA1c, glycoalbumin, blood glucose, triglycerides, creatinine, plasma pentosidine, and diastolic blood pressure.
I congratulate the authors for their work which supports the clinical utility of HRV.
The manuscript is of good quality, the design of the study is adequate, the results are correctly presented and the discussions are pertinent.
I have just two issues that need to be addressed:
- Throughout the paper, it is stated that reduced CVRR or SDNN were associated with different parameters, but I did not find in the text how the authors defined reduced CVRR / SDNN. What was the threshold for normal values? As I understand the statistics done, they were compared as continuous parameters. Could the authors clarify this issue?
- The values presented for SDNN mean and standard deviation seem extreme. Could the authors recheck the values? Usually, SDNN should be around 100 ms.
Author Response
Thank you for your review.
1. SDNN and CVRR were calculated as the standard deviation of the RR interval and SDNN / mean value of the RR interval×100, respectively, during the measurement. Although thresholds for SDNN and CVRR have been reported for mortality in patients with depressed left ventricular function after an acute myocardial infarction and assessment of diabetic autonomic neuropathy, we currently do not have sufficient information to identify the cutoff values for the HRV indices. Therefore, in this study, they were compared as continuous parameters.2. SDNN had been multiplied by 100, which has been corrected. Thank you very much for pointing this out.
Reviewer 3 Report
This study reports a cross-sectional study of members of the general population of Japan, non randomly selected, and correlates heart rate variability with other biometrics and demographics. The work seems mostly fine, though I’ve some suggestions for improvement prior to publication.
The results section is extremely sparsely written. I really think it would be improved by incorporating substantially more textual narrative about the findings, especially if the narrative can link the tables together. As it stands, the entire section is nine sentences that pretty much just state the next table is X and the significant variables are ABC. This leaves the reader having to find his or her own way through the tables and findings therein.
I personally would prefer if there were also a graphical representation of the data to afford the reader a better sense of what the relationships are like than the tables provide.
Description of units was often incomplete. For instance, L72 does not fully describe how BMI was calculated (since both mass and height are lacking units), while tables 2 to 4 lack units for both x and y variables. Since units are required to understand the beta coefficients, these absolutely must be included in the revised manuscript.
I would like to see more detail or a citation for the ‘a hill climbing method’ and the ‘a low pass filter’ (L96,98).
Table 1: Not clear why more precision was given for the SDs than for the means; surely it should be the other way around, or equal precision? (My preference would be to reduce the precision of the SDs to match the means. I would also invite the authors to consider rounding the %ages a bit more boldly.) I think there is a typo in the units for SDNN. Is there a reason to put the units and SDs in parenthesis? If not, this serves to make the table more visually messy than it need be, so consider removing the parentheses from these columns.
Consider providing anonymised data in a supplementary appendix if the ethics approval permits it. This would be preferable to having interested readers approach your ethics board for access. If it’s not possible, then may I suggest you consider explicitly baking this into your future studies?
Author Response
Thank you for your review.
- We have added to details of the analysis in the results.
- Units were added to the table of analysis results.
- We have added citations to ‘a hill climbing method’ and ‘a low pass filter’.
- Mean and SD accuracy are the same. I removed the parentheses for the unit display; I think it would be better to have parentheses for SD.
- Unfortunately, we are unable to release data for this study without the approval of the Ethics Committee.
We hope that these revisions are sufficient to make our manuscript suitable for publication in the healthcare and look forward to hearing from you at your convenience.
Round 2
Reviewer 2 Report
I thank the authors for addressing the issues I suggested.
The manuscript has been sufficiently improved to warrant publication.
Author Response
Thank you for your reviews.
Thanks to that, it became a better manuscript.
Reviewer 3 Report
Thank you for making the changes requested in my previous review. The new results section is not an improvement, however. It effectively just copies the results of the tables into the text without attempting any explanation. I was rather hoping more for some words to guide the reader through your results, something like "The largest associations we found with Y were between X1, X2 and X3. There was a strong, positive increase with X1, with Y increasing by about ZZZ units per unit increase in X1, and smaller associations with etc etc. A statistically significant association was also discerned with X4, but the magnitude was small (ZZZ) and thus of marginal clinical significance" or similar. Dumping the table into the text without filtering it at all seems slightly disrespectful to the reader.
Author Response
Thank you for pointing this out, it became a better manuscript.
I consulted with other authors with PhD and modified the way the results were described.
We would appreciate it if you could check our revised manuscript highlighted in yellow.